# Topographic DCNNs trained on a single self-supervised task capture the functional organization of cortex into visual processing streams

**Dawn Finzi**
Stanford University
dfinzi@stanford.edu

**Eshed Margalit**
Stanford University
eshedm@stanford.edu

**Kendrick Kay**[*]
University of Minnesota
kay@umn.edu

**Daniel L. K. Yamins**[*]
Stanford University
yamins@stanford.edu

**Kalanit Grill-Spector**[*]
Stanford University
kalanit@stanford.edu

## Abstract

A key organizing principle of visual cortex is functional specialization, whether locally in the context of category-selective patches, or on a broader scale in the case of visual processing streams. Primate visual cortex has traditionally been divided into two such processing streams, though recent research suggests that there may be at least three functionally and anatomically distinct streams, extending along the ventral, lateral, and parietal surfaces of the brain. While processing streams are typically thought of within the framework of what downstream behaviors/tasks they support, here we ask instead whether anatomical constraints may be sufficient to produce this differentiation, even within the context of just one task objective. Comparing directly to human fMRI responses, we show that a model trained on a single task, and with novel anatomical constraints (Topographic DCNN), can capture not only the functional responses but also the segregation of visual cortex into distinct processing streams. The match to human data is strongest for a self-supervised vs. supervised objective and when the anatomical constraint, which encourages local response correlations as proxy for minimizing wiring length, is appropriately weighted. These results suggest that the broad-scale functional organization of visual cortex into parallel processing streams may be explained by the pressure to minimize biophysical costs such as wiring length, and that local spatial constraints can surprisingly percolate to create broad-scale structure.

## 1 Introduction

A third of the human brain is devoted to vision, with tasks such as visually perceiving an object's category, dynamics, and location crucial to interacting with and acting upon on our world. Traditionally, visual cortex has been divided into two functionally and anatomically distinct processing streams: a "what" stream extending along ventral occipito-temporal cortex, and a "where" or "visually-guided grasping" dorsal stream extending along occipito-parietal cortex[1, 2]. More recently, a third lateral stream, extending along lateral occipito-temporal cortex, has been proposed, with suggested functions including multimodal processing[3], action recognition[4], and social perception[5]. However, it is an open question why cortex might organize into streams in the first place. Are different tasks required for generating different streams (prevailing hypothesis) or are anatomical constraints sufficient to produce multiple streams even in the context of a single task objective?

---

[*]Co-senior author.

4th Workshop on Shared Visual Representations in Human and Machine Visual Intelligence (SVRHM) at the Neural Information Processing Systems (NeurIPS) conference 2022. New Orleans.

One such potential anatomical constraint is wiring length minimization, i.e., an architectural bias toward short-range connections[6, 7]. This principle has been proposed to explain the functional topography of orientation preference in primary visual cortex[8] as well as category-selective patches in high-level visual cortex[9, 10]. Recently, a new model class has been introduced, Topographic Deep Convolutional Neural Networks (DCNNs)[9, 11], which encourages local correlations as a proxy for wiring length minimization. Using an end-to-end optimization framework that minimizes combined task and spatial loss terms, Topographic DCNNs both well-predict responses in the ventral stream as well as capture the functional topography of V1 and ventral temporal cortex (VTC)[11]. Here, we leverage Topographic DCNNs to test if they (1) predict responses to many stimuli across not just ventral, but also lateral and parietal cortex and (2) recapitulate the broad-scale functional topography of multiple visual processing streams. The outcome of these tests will provide insights into what constraints are sufficient to reproduce this organization (see A.1 for Related Work).

## 2 Methods

### 2.1 Topographic DCNNs

As a candidate model class we use Topographic DCNNs, with ResNet-18 as an architectural backbone. Each convolutional layer in the network is converted to a topographic layer by assigning each model unit to a position in a two-dimensional simulated cortical sheet that is unique to that layer. For the purposes of comparing to higher-level visual cortex, we will be focusing on the final convolutional layer, which has been previously shown to best match primate central IT and human VTC [12–14]. Model positions for each layer are initialized to be coarsely retinotopic, shuffled in small neighborhoods, and then permanently fixed to those positions; network parameters are randomly initialized.

Models are trained to minimize a composite loss function with two components: (1) a task component: SimCLR cross-entropy (CE) loss[15] and (2) a spatial loss (SL) component that encourages nearby model units to be correlated in their responses, computed on each batch as follows:

$$\text{Loss} = \text{CE} + \sum_{l \in \text{layers}} \alpha_l \text{SL}_l \tag{1}$$

CE is the cross-entropy task loss component, $N$ is the number of convolutional layers in the model, $\alpha_l$ is the weight of the spatial loss component (fixed across all layers), and $\text{SL}_l$ is the spatial correlation loss computed for the $l$-th layer. Specifically, $\text{SL}_l$ is computed on a given batch by randomly sampling a local neighborhood and calculating for pairs of units, (1) Pearson's $r$ between the response profiles, ($\overrightarrow{C}$), and (2) the vector of physical (position) similarity ($\overrightarrow{D}$), specifically the stabilized reciprocal Euclidean distance:

$$\overrightarrow{D} = \frac{1}{(1 + \overrightarrow{d})} \tag{2}$$

where $\overrightarrow{d}$ is the vector of pairwise cortical distances. These two terms are then related as follows:

$$\text{SL}_l = 1 - \text{Corr}\left(\overrightarrow{C}, \overrightarrow{D}\right) \tag{3}$$

such that $\text{SL}_l$ is minimized when nearby units have correlated responses to the training stimuli.

We compare models across two factors: degree of spatial constraint and training task. The spatial constraint varied from $\alpha = 0$, yielding a nonspatial model in which unit positions are ignored during training, to $\alpha = 25$, a weighting of the spatial loss function such that local correlations dominate over task. We used two training tasks: (i) self-supervised contrastive learning (SimCLR[15]), which increases the similarity of representations between different augmentations of the same image while decreasing the similarity to different images. This type of representation structure may be beneficial for a range of downstream functions associated with any of the three streams. And (ii) supervised object categorization. This task has been shown to best predict responses in the primate ventral stream[16, 17] and can serve as a benchmark. All models were trained using ImageNet[18].

### 2.2 Neural data

As our neural comparison, we used the Natural Scenes Dataset (NSD)[19], a high-resolution fMRI dataset that densely sampled responses to up to 10,000 natural images in each of eight individuals

(see Allen et al. [19] and A.2 for details on preprocessing of the data). We defined regions of interest (ROIs) for early, intermediate, and high-level visual cortex for each of the three streams based on a combination of anatomical landmarks, noise ceiling estimates, and a constraint to roughly match the number of voxels per stream. We focus only on the high-level ROIs for the purposes of this paper. Single-trial responses were z-scored across images for each voxel and session and then averaged across 3 trial repeats. For comparisons between models and individual brains, we leverage the full set of images (6234-10,000 per individual). For comparisons between individuals, we used 515 shared images which all participants viewed 3 times. We used cortex-based alignment to align all data to the fsaverage surface, and report here results from the right hemisphere.

## 2.3 Mapping between models and brains

As traditionally used linear encoding[16] and representational similarity analysis[17] methods deliberately re-weight or marginalize the outputs across units, thus obscuring topographic organization, we developed new mapping algorithms to compare model and brain functional topography. We used a direct mapping (1-to-1) between model units and brain voxels, mapping each unit to a single voxel. As a benchmark, we tested the new mapping algorithms on how well they mapped voxels from one brain to another (brain-to-brain case). In the brain-to-brain case, we computed the correlations between responses to the shared images for each pair of source and target voxels in different brains (same procedure in the model-to-brain case using activations of model units in response to the images each subject viewed). The goal was to determine from the vast space of possibilities, the optimal assignment of each source voxel (or unit) to a single corresponding target voxel. We first computed the assignment solely on the basis of functional similarity (Alg. 1), using the Kuhn-Munkres algorithm[20]. We then incorporated our prior for some level of smoothness in the mapping by implementing an iterative version of the Kuhn-Munkres algorithm that gently encourages neighboring voxels in the target space to "pick" neighboring voxels in the source space (Alg. 2; see A.3).

## 3 Results

### 3.1 1-to-1 voxel mapping based on response similarity alone only partially recovers the structure of processing streams

We began by mapping one subject's brain to another subject's brain using just functional similarity (Alg. 1). We examined the resulting mapping by coloring each voxel in the target individual's brain by which stream it came from in the source individual's brain (Fig. 1A, example subject pair). This revealed that voxels from one stream in the source brain were largely assigned to the same stream in the target brain. However, there were also many mismatches. To quantify the degree of correspondence, we calculated what percentage of voxels from each target brain and ROI were assigned to voxels from the same ROI in the source brain. Correspondence was significantly above chance for each stream (chance=33%; mean $\pm$ SE: Ventral = $56.5\pm0.7\%, p = 9.1\text{x}10^{-9}$; Lateral = $62.6\pm0.7\%, p = 2.2\text{x}10^{-9}$; Parietal = $58.8\pm0.4\%, p = 1.2\text{x}10^{-10}$;

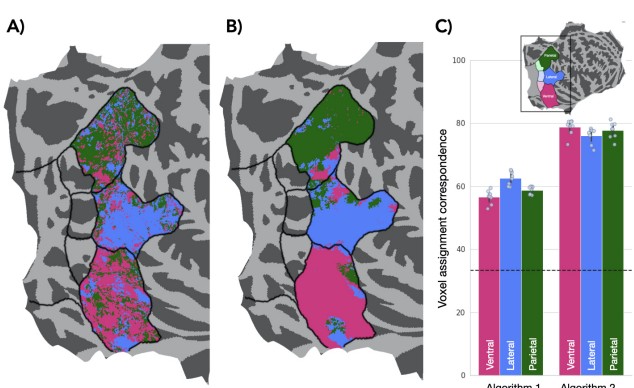

Figure 1: (A) Voxels in Target (Subj. 2) brain colored by their assignment to Source (Subj. 1) brain using an algorithm that matches voxels solely on the basis of functional similarity (Alg. 1). (B) Voxels in Target (Subj. 2) brain colored by their assignment to Source (Subj. 1) brain using an algorithm (Alg. 2) that additionally incorporates a gentle smoothness constraint (25 iterations). (C) Top right: zoomed region in (A,B) shown on full flat map. Bars: Comparison of voxel-to-voxel correspondence. Data plotted for high-level ROIs across three streams, averaged across source subjects for each target subject. Color represents stream throughout manuscript (pink: Ventral, blue: Lateral, and green: Parietal). Dotted line: chance level (33%). Error bar: 95% CI, each dot is a subject.

Fig. 1C). We then tested if incorporating a smoothness constraint (Alg 2) enhances the mapping. Results indicate significantly improved brain-to-brain correspondence for Alg. 2 vs. 1 ($F(1,7) = 670.9, p = 3.3\text{x}10^{-8}$, Fig. 1B, example subject pair) yielding an average of 78% correspondence (Fig. 1C). Thus, we used Alg 2. to map model units to brain voxels.

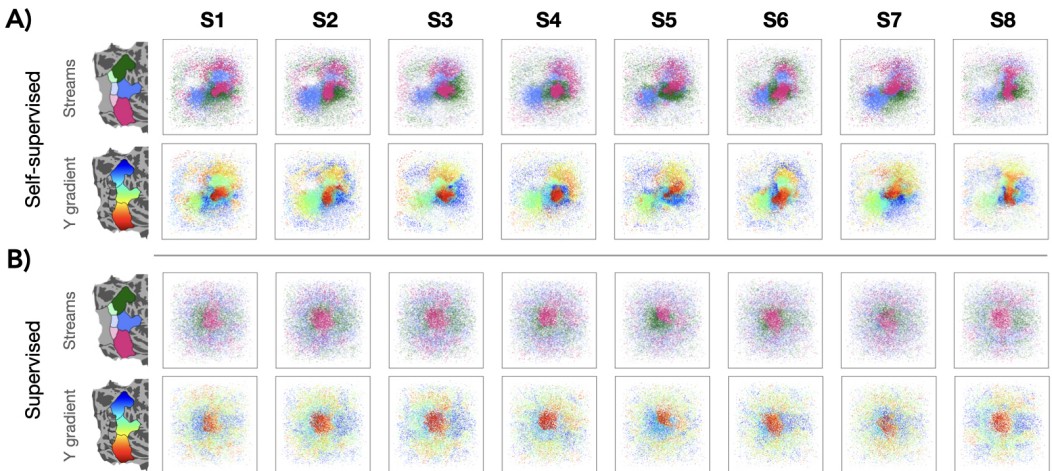

Figure 2: Model-to-brain mapping for a model trained on ImageNet and $\alpha = 0.25$ using a self-supervised, SimCLR (A) or supervised, object categorization (B), objective. Each panel shows the assignment of model units in the simulated cortex of the last convolutional layer to a voxel in a target brain. Each dot is a model unit. Units in the simulated cortex are colored by stream (top) or superior-to-inferior gradient (y-position in flat map, bottom). Each column displays the model-to-brain mapping for one individual. Opacity of the units reflects the strength of the unit-to-voxel correlation between responses to NSD images).

## 3.2 Topographic DCNNs can capture the broad-scale functional topography of high-level visual cortex

We next compared how well model units mapped to voxels in individual brains. We tested the assignment and topographic organization of units in the last convolutional layer vis-a-vis voxels in high-level ROIs of the Ventral, Lateral, and Parietal streams. Specifically, we asked how good the functional correspondence would be between model units and voxels and whether there would be clustering of units by stream in the simulated cortex. Results in Fig. 2A demonstrate that for Topographic DCNN models trained on SimCLR with a spatial loss weighting of $\alpha = 0.25$, unit assignments are clustered by stream in the simulated cortex. This correspondence is highly consistent across subjects. Critically, this does not depend on our ROI definitions or deliberate division into three streams, as this correspondence also emerges when visualizing unit assignments along a superior-to-inferior axis (y-gradient, on flat map). In contrast, training with the same architecture and spatial loss weighting of $\alpha = 0.25$ but on a supervised categorization task does not yield this three stream structure. Instead, this model develops units that are assigned to and somewhat clustered for the Ventral stream (center of the model simulated sheet, Fig. 2B), but not for the Lateral or Parietal streams.

## 3.3 Broad-scale functional topography is only captured by self-supervised, not supervised, Topographic DCNN, and an optimal weighting of the spatial loss term

The recapitulation of the three-stream functional organization of the brain that we see in the SimCLR-trained $\alpha = 0.25$ model is dependent on particular parameters, as it does not emerge in a number of other settings. First, the mapping algorithm used does not guarantee this result, as we can see in an instantiation of a SimCLR-trained Topographic DCNN with $\alpha = 0$ (essentially a vanilla ResNet-18). While the unit-to-voxel assignments in this model have some local structure (Fig. 3A, left panel), the functional topography is qualitatively different than the same task loss with a spatial loss weight of $\alpha = 0.25$; crucially, it does not capture the three-stream organization of the brain. In fact, the qualitative correspondence varies with the degree of spatial weight, steadily increasing from $\alpha = 0$ to $\alpha = 0.25$ and then deteriorating for $\alpha > 1$ as the spatial loss dwarfs the task loss (Fig. 3A). We find a similar pattern if unit assignments are by position along a superior-to-inferior axis (Fig. 3B).

We next quantified the unit-to-voxel correspondence via two metrics: (i) the number of clusters per stream ROI calculated using an automatic cluster detection algorithm[11], and (ii) the "clusteriness", measured by calculating entropy of the 2D histogram for ROI assignment in comparison to a randomly shuffled layout (details in A.6). For both metrics we compared the model-to-brain mapping to the brain-to-brain mapping. The three-stream model of the brain predicts one cluster per stream ROI. Automated cluster detection revealed that there is a narrow range of that provide this match across

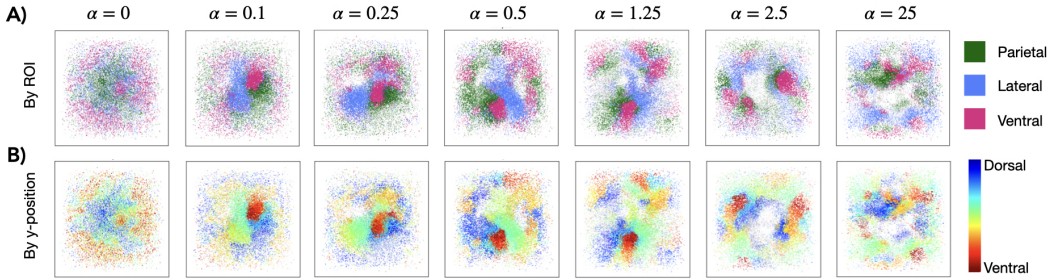

Figure 3: Model-to-brain mappings across a range of spatial weights for an example subject, colored by (A) stream ROI assignment or (B) superior-to-inferior gradient on cortex (y-position). Each dot is a model unit. See A.4 for all subjects.

streams for the self-supervised models (Fig. 4A). For $\alpha = 0.25$, the average is 1.375 (SE=0.17) clusters for Ventral, 1.5 (SE=0.18) clusters for Lateral, and 1.375 (SE=0.43) clusters for Parietal. For this $\alpha$, the average number of clusters did not significantly differ from one for any of the three streams (Wilcoxon signed-rank test, Bonferroni corrected for 3 comparisons, all $ps > .14$). Likewise, estimated entropy was most similar to brain-to-brain mappings for $0.25 \geq \alpha \leq 0.5$ (Fig. 4B), and for $\alpha = 0.25$ did not significantly differ from the brain-to-brain mapping for Ventral or Lateral, though it did for Parietal ($t(7) = 4.95, p = 0.005$, Bonferroni corrected for 3 comparisons). Overall, quantitative measures of correspondence were better for Ventral and Lateral than Parietal, indicating an area for improvement.

Mirroring what we observed qualitatively, the Topographic DCNN models trained on the supervised task of object categorization, show a worse match to the brain on the quantitative metrics. Only at the highest spatial weights, by which point the functional correspondence has completely broken down (Fig. 4E), do we see the number of patches detected close to one per stream, or the histogram entropy near the brain-to-brain mapping. At $\alpha = 0.5$, the supervised model with the highest functional correspondence, we see a very clear Ventral cluster, (see A.5 for model-to-brain maps for all spatial weights for the supervised models) and in fact the histogram entropy for Ventral is significantly lower than the brain-to-brain case (model-to-brain mean±SE = 0.83±.008, brain-to-brain mean±SE = 0.89±.004, $t(7) = -7.7, p = 0.0003$, Bonferroni corrected for 3 comparisons, Lateral and Parietal model-to-brain values significantly *higher* than brain-to-brain values, all $ps < .002$), indicating an even greater degree of clustering for the Ventral stream than found in the brain, with minimal clustering for the other two streams.

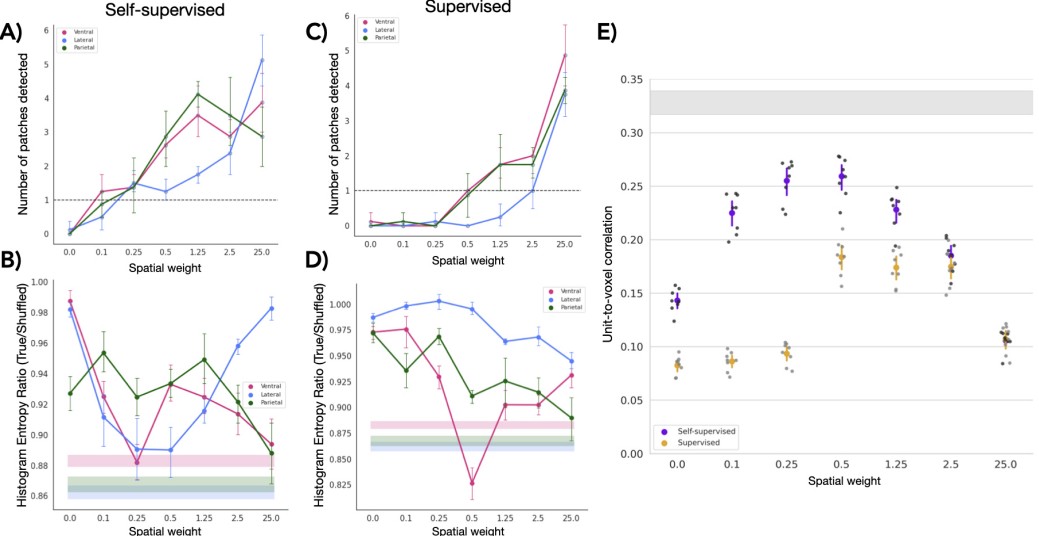

Figure 4: (A) Number of clusters in model-to-brain mappings by spatial weight for the self-supervised models. Dotted line: the number of clusters per stream observed in the brain. (B) "Clusteriness" as measured by 2D histogram entropy for the model-to-brain mapping for the self-supervised models. Shaded bars: same except for brain-to-brain mapping (mean±SE across target subjects). (C) Number of clusters for the supervised models. (D) "Clusteriness" as measured by 2D histogram entropy for the supervised models. (E) The average correlation between assigned unit and voxel responses on a left-out test set, normalized by each voxel's noise ceiling. Each dot represents a subject. Shaded gray bar: same calculation for brain-to-brain mapping (mean±SE across target subjects).

As referenced above, we also quantified the functional correspondence between unit and voxel responses to the NSD images as a function of $\alpha$ and training task, using the cross-validated unit-to-voxel correlations. Strikingly, unit-to-voxel correlations were significantly higher for the Topographic DCNNs trained with the more biologically plausible self-supervised objective vs. the supervised task, particularly at lower spatial weights (repeated measures ANOVA, significant main effects of task $F(1,7) = 2070.6, p = 6.5\text{x}10^{-10}$, and spatial weight, $F(6,42) = 499.7, p = 2.2\text{x}10^{-37}$; task x spatial weight interaction, $F(6,42) = 418.4, p = 8.5\text{x}10^{-36}$), and at no point do the correlations for supervised models approach the correlation of the best self-supervised model (mean$\pm$SE = 0.26$\pm$.006). The unit-to-voxel functional correlation varies with spatial weight, tracking the pattern we observed for the topographic correspondence. That is, spatial weights that yielded a better topographic correspondence between model and brain also yielded a better functional correspondence (Fig. 4E). This indicates that the spatial loss does not only change the spatial layout, but also meaningfully changes the functional tuning of model units.

## 4 Discussion

Using a newly developed Topographic DCNN and novel model-to-brain mapping method, we show that it is possible to capture much of the broad-scale organization of visual cortex into processing streams in a model trained on a single self-supervised task, with the addition of a spatial constraint. This suggests that the functional organization of cortex into distinct visual processing streams could emerge simply from the biophysical pressure to minimize wiring length, while learning generally useful visual representations. It is important to note that the spatial loss used here does not explicitly impose a wiring length cost. However, the consequence of nearby units having similar responses is that wiring length is minimized[11] and related work has shown that explicitly minimizing wiring length increases the functional coupling between nearby units[10].

Of note, the structure we see in the topographic models does not fully match macroscale structure of visual cortex, in that the Parietal clusters are not as clearly formed, and consistently emerge near Ventral clusters, whereas in the brain we see a progression from Ventral to Lateral and then to Parietal cortex. This suggests an area for improvement. Interestingly, this also seems to reflect the greater similarly of Ventral and Parietal representations (vs. Lateral and Parietal) in this dataset [14]. We, very speculatively, note that there is a white-matter fiber bundle directly connecting Ventral and Parietal (the Vertical Occipital Fasciculus) but no such direct connection between Lateral and Parietal. We hope to explore the role of long-range white-matter connections in the future.

Finally, given the dominant role task has played in the field's thinking about visual processing streams, a logical question is whether using different training tasks would better differentiate the streams. We tested this by training three networks with different tasks commonly proposed as the functional objective of each stream (Ventral: object categorization, Lateral: action recognition, and Parietal: object detection; ResNet-50 base architectures). Consistent with prior work using linear regression to map units to brain responses [14], we find that models trained on these three tasks minimally differentiated the streams (see A.7). Crucially, however, this does not preclude the spatial constraints *creating* functional differentiation. While spatial constraints may dictate a particular way to physically layout neurons capturing a range of representations, groups of these neurons may also then be more or less useful for different downstream tasks. In fact, we hope to test this in future work by learning readout layers for units assigned to each of the three streams separately and comparing performance on different tasks.

In conclusion, our results suggest a rethinking of visual processing streams and highlight the crucial role of anatomical constraints, above and beyond task demands, in their emergence.

## Acknowledgments and Disclosure of Funding

This work was supported by NIH grant to R01EY023915 to K.G.S. Collection of the NSD dataset was supported by NSF IIS-1822683 and NSF IIS-1822929.

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

# A  Appendix

## A.1  Related work

**Models of the topographic organization of ventral cortex.**   Recent work has moved beyond self-organizing maps to image-computable models which attempt to capture both the response properties and topographic organization of VTC. In particular, three DCNN-based approaches, the Interactive Topographic Network (ITN)[10], the "VTC-SOM"[21], and the Topographic VAE[22] successfully produce maps with clustered category-selectivity as found in VTC. Our approach varies from prior ones in that Topographic DCNNs are trained end-to-end, allowing us to investigate how the training impacts the learned kernels, their fit to brain responses, *and* their topographic organization.

**DCNN models of the lateral and parietal streams.**   DCNNs trained on supervised object recognition[16, 17, 23] and contrastive self-supervised objectives[12]) are excellent predictive models of responses across the ventral stream. However, few studies have used DCNNs to predict responses in lateral and parietal cortex, with the following exceptions: DCNNs trained for action recognition in videos[24] or for predicting an agent's self-motion[25] well-predict responses in V3A/B, MT and MST[2]. Additionally, one study attempted to model both the ventral and dorsal pathways using a branched architecture trained with a self-supervised predictive loss function[26], though in mice and without matching to the topography.

## A.2  Additional information on drawing of ROIs and processing of neural data

A full set of seven (one early, three intermediate and three higher-level) ROIs were drawn on the fsaverage surface as follows:

**Early visual cortex ROI:** The early visual cortex ROI was drawn as the union of the V1v, V1d, V2v, V2d, V3v and V3d ROIs from the Wang retinotopic atlas. Additionally, V2v and V2d were connected such that the part of the occipital pole typically containing foveal representations was also included. The same was repeated for V3v and V3d.

**Intermediate ROIs:** Three intermediate ROIs were drawn corresponding to each of the three streams: ventral, lateral and parietal. All three ROIs border the early visual cortex ROI on the posterior side. The intermediate ventral ROI was drawn to reflect the inferior boundary of hV4 from the Wang atlas and to include the inferior occipital gyrus (IOG), with the anterior border of the ROI drawn based on the anterior edge of the inferior occipital sulcus (IOS). The intermediate lateral ROI was drawn directly superior to the intermediate ventral ROI, with the superior and anterior borders determined as the LO1 and LO2 boundaries from the Wang atlas. The intermediate parietal ROI was drawn directly superior to that, reflector exactly the borders of the union of V3A and V3B from the Wang retinotopic atlas.

**Higher-level ROIs:** Three higher-level ROIs were drawn for each of the Ventral, Lateral and Parietal streams, bordering their respective intermediate ROIs on their posterior edges. The ventral ROI was drawn to follow the anterior lingual sulcus (ALS), including the anterior lingual gyrus (ALG) on its inferior border and to follow the inferior lip of the inferior temporal sulcus (ITS) on its superior border. The anterior border was drawn based on the midpoint of the occipital temporal sulcus (OTS). The lateral ROI was drawn such that the higher-level ventral ROI was its inferior border and the superior lip of the superior temporal sulcus (STS) was used to mark the anterior/superior boundary. The rest of the superior boundary traced the edge of angular gyrus, up to the tip of the posterior STS (pSTS). The parietal ROI was drawn to reflect the boundary of the lateral ROI on its inferior edge and to otherwise trace the borders of and include the union of IPS0, IPS1, IPS2, IPS3, IPS4, IPS5 and SPL1 from the Wang retinotopic atlas.

The three higher-level ROI were then trimmed using the prepared noise ceiling maps for beta version b3 and the fsaverage surface[19]. The noise ceiling estimates represent the amount of variance contributed by the signal expressed as a percentage of the total amount of variance in the data, for the average of responses across three trial presentations. An approximate cutoff of 10% was used to guide trimming of the higher-level ROIs, such that we were left with reduced ROIs where all

---

[2]Of note, while MT is traditionally thought of as within the dorsal stream, including in the cited studies, anatomically, MT is located within the lateral stream, not the parietal stream. This distinction is particularly apparent in humans, where the lateral surface is greatly expanded in comparison to non-human primates.

voxels had a noise ceiling $\geq 10\%$ theoretically predictable variance. These ROIs were contiguous and roughly matched in size (right hemisphere number of voxels per ROI: Ventral = 5638, Lateral = 6839, Parietal = 6688). In the main text, only the higher-level ROIs are used for analysis. However, the full details for drawing all 7 ROIs are included here for completeness and as the higher-level ROIs using the boundaries of the intermediate ROIs as their posterior borders.

Additionally, note that for the purposes of this paper, we use refer to the brain units of measurement as "voxels", though they are more technically "vertices" as we are using the fsaverage preparation.

## A.3 Full algorithm in pseudo-code for Algorithm 2

---

**Algorithm 2** One-to-one mapping including a smoothness constraint

---

**Require:** Cost matrix $C$ of dimension $N_s \times N_t$, where $N_s$ is the number of source units and $N_t$ is the number of target units, and each entry in $C$ is $1-$ the pairwise correlation of the response vectors. Source distance matrix, $D_s$, of dimension $N_s \times N_s$, with the pairwise distances between all units in the source space. Target distance matrix, $D_t$, of dimension $N_t \times N_t$, with the pairwise distances between all units in the target space. Radius $r$ to use as neighborhood size.
    **procedure** ALGORITHM2($C, D_s, D_t, r$)
        Assignments $A \leftarrow$ Kuhn-Munkres($C$)       ▷ Initialized based on purely functional mapping
        **while** Mean movement of assignments from iteration to iteration has not converged **do**
            $C_{\text{temp}} \leftarrow C$.copy()
            **for** each target unit, $v_t$ **do**
                Find all neighboring target units, $V_{tn}$ within distance $r$
                Initialize candidate matrix, $V_{sn}$
                **for** each unit in $V_{tn}$ **do**
                    Find their assigned source unit, $v_s$, from $A$
                    $v_{sn} \leftarrow$ all units in source space within distance $r$ from $v_s$
                    Append $v_s$ and $v_{sn}$ to $V_{sn}$
                **end for**
                Fit a 2D Gaussian, $G$, to point cloud, $V_{sn}$ in source space
                $M_{sn} \leftarrow$ mahalanobis($V_{sn}, G$)
                **for** each unit, $u$, in $V_{sn}$ **do**
                    **if** $M_{sn}[u] > 2$ **then**
                        Remove $u$ from $V_{sn}$
                    **end if**
                **end for**
                $C_{\text{temp}}[v_t, \neg V_{sn}] = 1000$ ▷ all source units not in $V_{sn}$ are set to have a prohibitive cost
            **end for**
             $A \leftarrow$ Kuhn-Munkres($C_{\text{temp}}$)
        **end while**
        **return** $A$
    **end procedure**

---

For the brain-to-brain case, radius $r$ was set to 5mm. To convert this radius to the model space, which is $10 \times 10$, we calculated what percentage of the max voxel-to-voxel distance (237 mm) the brain radius cutoff was and then multiplied that percentage by the max model distance (12.9). This yielded a model radius cutoff of approximately .27. Thus, in the model-to-brain case two radii are used, 5mm for the brain distances and .27 for the model distances.

## A.4 Model-to-brain maps for all subjects and spatial weights

See figures S1, S2, S3, S4, S5, S6.

## A.5 Model-to-brain maps for all spatial weights for the supervised models for an example subject

See figures S7

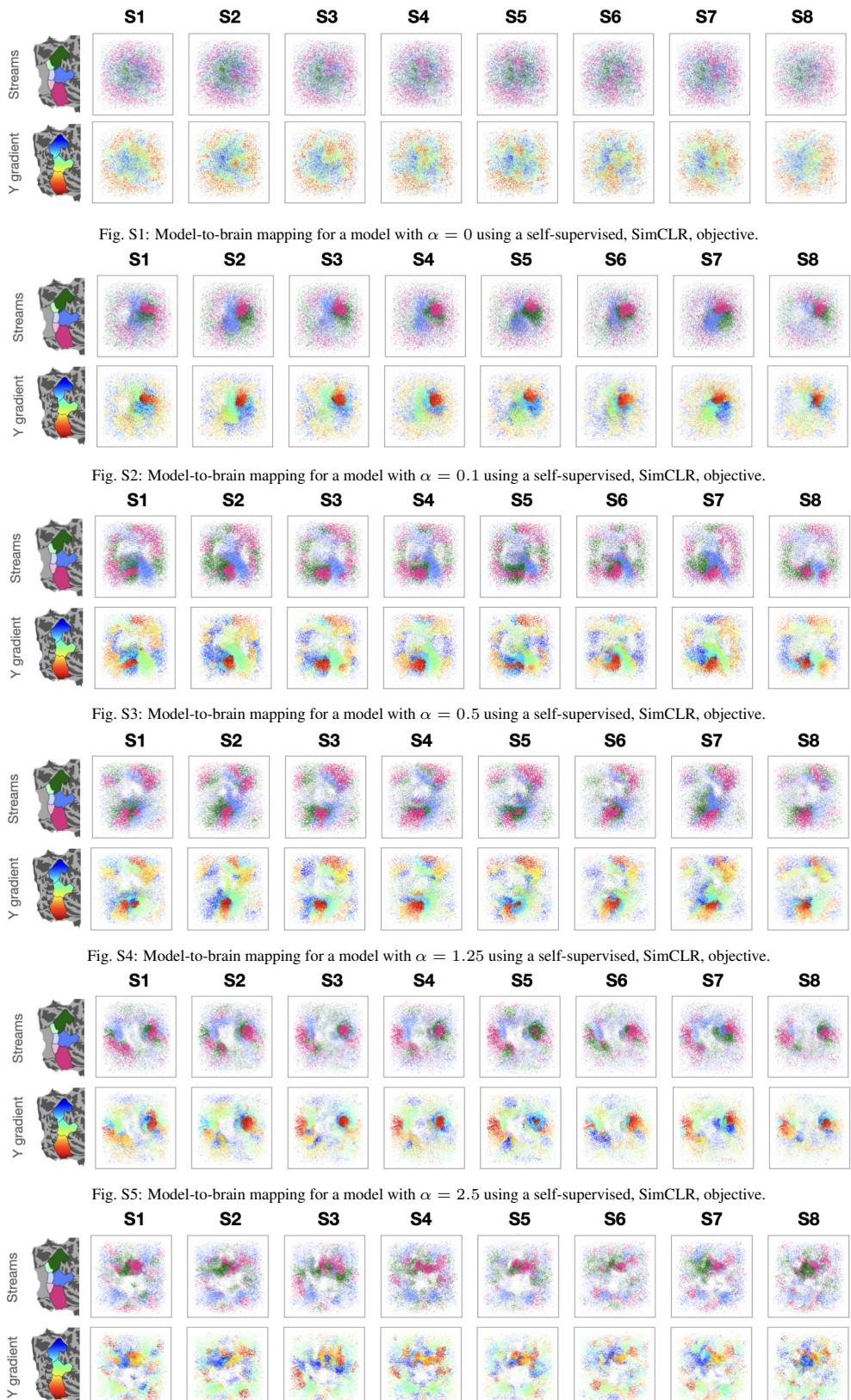

Fig. S1: Model-to-brain mapping for a model with $\alpha = 0$ using a self-supervised, SimCLR, objective.

Fig. S2: Model-to-brain mapping for a model with $\alpha = 0.1$ using a self-supervised, SimCLR, objective.

Fig. S3: Model-to-brain mapping for a model with $\alpha = 0.5$ using a self-supervised, SimCLR, objective.

Fig. S4: Model-to-brain mapping for a model with $\alpha = 1.25$ using a self-supervised, SimCLR, objective.

Fig. S5: Model-to-brain mapping for a model with $\alpha = 2.5$ using a self-supervised, SimCLR, objective.

Fig. S6: Model-to-brain mapping for a model with $\alpha = 25$ using a self-supervised, SimCLR, objective.

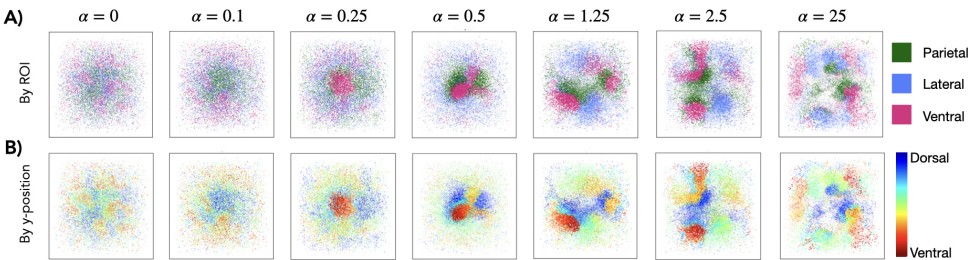

Fig. S7: Model-to-brain mappings for the supervised models across a range of spatial weights for an example subject, colored by (A) stream ROI assignment or (B) superior-to-inferior gradient on cortex (y-position).

## A.6 Additional information on measures of "clusteriness"

Two measures of "clusteriness" are included within the main text. First, we calculated the number of clusters per stream using an automatic cluster detection algorithm [11]. Model units on the model cortical sheet were assigned stream labels based on the location of the each unit's assigned voxel in the subject brain, and then these maps were divided based on stream assignment, yielding three separate maps (one for each of Ventral, Lateral and Parietal). These maps were smoothed with a two-dimensional Gaussian with $\sigma = 0.1$ and then thresholded at 50%. Groups of contiguous pixels that represented at least 5% of the total map were grouped and considered a cluster. This process was applied to all combinations of subjects and model to determine the number of clusters for each stream and mapping.

Secondly, we attempted to evaluate the level of spatial clustering among unit assignments using a histogram-based approach. In this approach, we once again divide each model-to-brain map into three binary maps by stream assignment. As a control, we also generate three maps for each model-to-brain mapping where, for each stream, we took the number of units assigned to that stream and created a binary map where the positions were randomly shuffled across possible unit positions. We then divided each map into a $N \times N$ grid, where $N$ was calculated as 10% of the total extent of the mapping (yielding a $10 \times 10$ grid of bins in the model-to-brain case), and calculated the entropy of the 2D histogram. We chose to calculate entropy of the 2D histogram as high entropy across bins would suggest high disorder/randomness of the point distribution while low entropy would signify a non-random distribution of points across spatial locations, i.e. "clusteriness". This value for the actual assignment (True) was normalized by the value for the randomized maps (Shuffled), such that a value of 1.0 indicates a model-to-brain mapping for that stream which was as randomly distributed as a random shuffle of the points (i.e. no "clusteriness") and lower values indicate a higher degree of "clusteriness". The same procedure was done for the subject-to-subject maps as a comparison point.

We also calculated a metric based on Ripley's $K$ [27], which is simply computed by centering a circle of radius $d$ on each sampling point and counting the number of neighboring events that fall inside it ($d$ was set to equal 10% of the total extent of the map, to mirror the 2D histogram entropy calculation). Ripley's $K$ was again calculated for the actual assignment (True) and normalized by the value for the randomized maps (Shuffled) for each stream and model-to-brain (or brain-to-brain) mapping. In

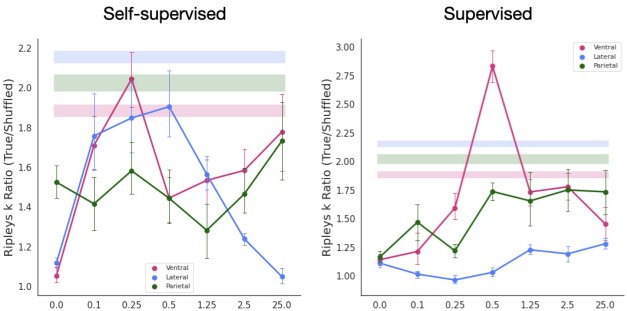

Fig. S8: "Clusteriness" as measured by Ripley's $K$ for the model-to-brain mapping for the self-supervised and supervised models. Shaded bars: same except for brain-to-brain mapping (mean±SE across target subjects).

this case, higher values indicate more "clusteriness". While exact values differ, results using this approach and the 2D histogram entropy approach largely converge (see S8).

### A.7    Task as an organizing force

#### A.7.1    1-to-1 mapping

We attempted to test the hypothesis that different tasks would serve as an equal or better organizing force in the following way. We chose models trained on object categorization, action recognition and object detection as they have been commonly proposed in the literature as candidate tasks for the Ventral, Lateral and Parietal streams respectively. For the object categorization model, we used a ResNet-50 trained on object categorization on ImageNet. For the action recognition model, we chose the SlowFast model architecture [28], which is a dual-pathway network with a 3D ResNet-50 backbone trained on the Kinetics-400 video dataset [29]. Finally, for the object detection model, we used a Faster R-CNN[30], with a ResNet-50 architecture as its base network, trained on MS-COCO [31]. For all networks, we used layer 4.1 or its equivalent, as this was found to be the best fitting layer for all of the high-level stream ROIs in previous results using linear regression [14]

We then subsampled an equal number of units from each network so that the total number of units was equal to the total number of voxels and then applied the one-to-one mapping procedure (Algorithm 1) from the main text to assign units to voxels (we were unable to apply Algorithm 2 as the units in these networks do not have spatial positions). For each stream ROI, we could then quantify what percentage of voxels "chose" what model. We additionally ran two control analyses: (1) a partially random control analysis where the object recognition model was trained but the action recognition and object detection models were not, in order to determine how much of the organization was mainly due to the organizing power of the object recognition model, and (2) a fully random control where random weights were used for all networks, and thus any structure in assignment could not be attributed to the task.

We found that voxels from each stream ROI did in fact "choose" units from models trained using their hypothesized task slightly more than chance (mean $\pm$ SE: ventral and object recognition = 41.9% $\pm$ 0.7; lateral and action recognition = 37.8% $\pm$ 0.4; parietal and object detection = 35.8% $\pm$ 0.6). However, this was also true for the partially random control. A two-way repeated measures ANOVA with the factors analysis type ('trained', 'partially random', 'fully random') and ROI/task comparison ('ventral/object recognition', 'lateral/action recognition', 'parietal/object detection') revealed a significant interaction ($F(4, 28) = 9.4$, $p = 6.0 \times 10^{-5}$), which post-hoc Tukey tests showed was driven by significant differences between trained and fully random for the ventral/object recognition ($p = .001$) and parietal/object detection ($p = .007$) pairings. All comparisons between trained and the partially random control were non-significant (all $ps > 0.21$). This suggests that at least the above chance assignment of lateral voxels to action recognition and parietal voxels to object detection could not stem from training on the task objective.

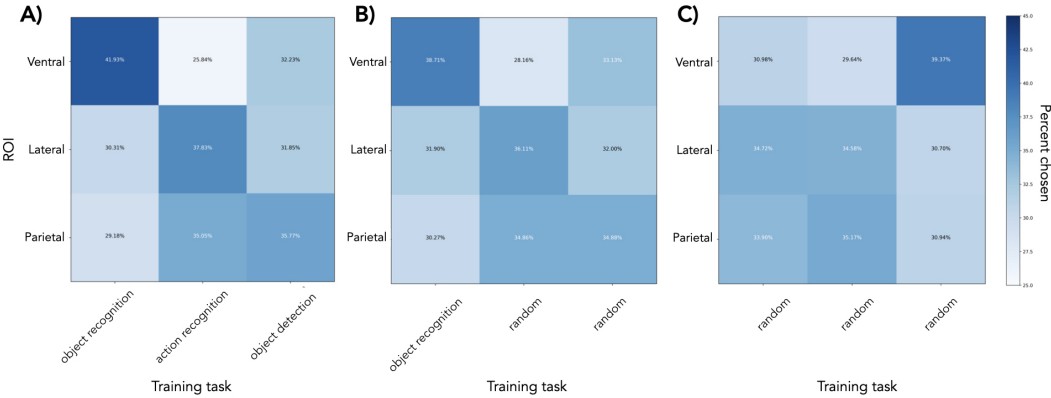

Fig. S9: For each row (i.e. stream ROI), the percentage of voxels from that ROI that "chose" each of the three candidate models (averaged across subjects). (A) Candidate models are 1. ResNet-50 trained on object recognition, 2. SlowFast trained on action recognition and 3. Faster-RCNN trained on object detection. (B) Partially random control where candidate models are 1. ResNet-50 trained on object recognition, 2. SlowFast untrained and 3. Faster-RCNN untrained. (C) Fully random control where candidate models are 1. ResNet-50 untrained, 2. SlowFast trained on action recognition and 3. Faster-RCNN trained on object detection.

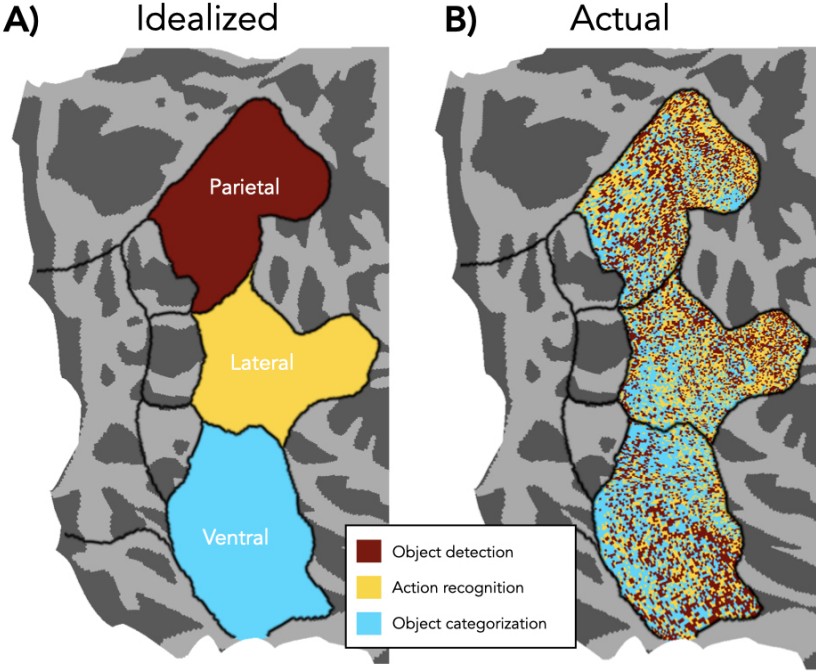

Fig. S10: Visualization of the task experiment. (A) Idealized results, i.e. what we would expect to see if all Parietal voxels were assigned to units from the object detection network, all Lateral voxels were assigned to action recognition and all Ventral voxels were assigned to object categorization. (B) Results from an example subject, illustrating that there is minimal structure in the assignments.

We also visualized these results on the flattened cortical surface by coloring voxels based on which network their assigned unit was from. If there was perfect correspondence between each of the streams and their hypothesized task, we would expect all the ventral voxels to be colored blue to indicate that they chose units from the network trained on object categorization, lateral yellow for action recognition and parietal red for object detection. Instead, we see a complete mix, with little discernable structure in the pattern of assignments (see S10 for an example subject).

### A.7.2 Linear regression results

We also used a linear encoding model approach, for comparison to earlier literature, where we extracted features in response to the NSD images for each layer of a number of object categorization and action recognition trained models. As in [13], we first projected these features into a lower dimensional space using a subsample of the ImageNet validation images and retained the first 1000 PCs. We then used ridge regression, with the alpha parameter cross-validated within the training set, to map these features to the individual voxel responses. Performance was evaluated on a left-out test set (80/20 split) for each subject separately and we report either the raw $R^2$ or the $R^2$ value normalized by the individual voxel noise ceiling for the best performing layer of each network. Subject-to-subject predictivity (shaded gray region) was calculated to match the model-to-subject mapping as closely as possible, using ridge regression, with N-1 subjects' voxel responses to the shared images for each ROI used as features to predict the left-out subject. This provides the noise ceiling.

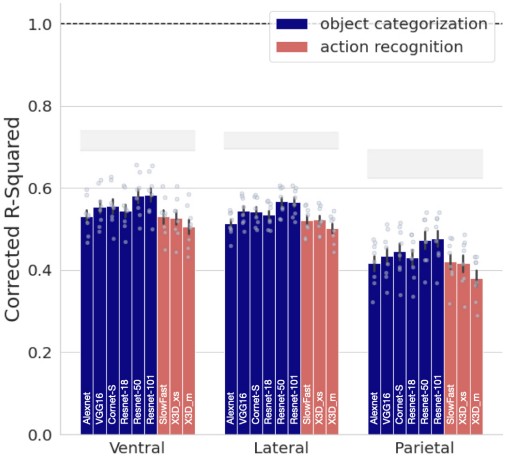

Fig. S11: Comparison of model fits across object and action recognition models & stream ROIs. $R^2$ values are normalized by the noise ceiling (NC) of each voxel such that 1.0 corresponds to the intrinsic data NC. Each dot represents a subject. Shaded gray error bars: range of subject-to-subject NC.

This preliminary analysis revealed that while some object categorization trained networks fit the ROIs better than others, these networks fit ventral and lateral (and parietal, when the subject-to-subject noise ceiling is taken into account), equally well. The same is true for the action recognition networks, as while they predict voxel responses across streams worse than the best object categorization trained models, their predictivity is similar across streams, particularly across the ventral and lateral ROIs (S11).

We also tested the performance of the topographic models using this methodology (S12). Unlike, the one-to-one mapping, this more flexible mapping did not separate out performance between the self-supervised and supervised models, though the self-supervised models still had a marginally higher $R^2$ at low spatial weights. Additionally, performance was comparable for the non-topographic ($\alpha = 0$) models and the topographic models, up until the highest spatial weights.

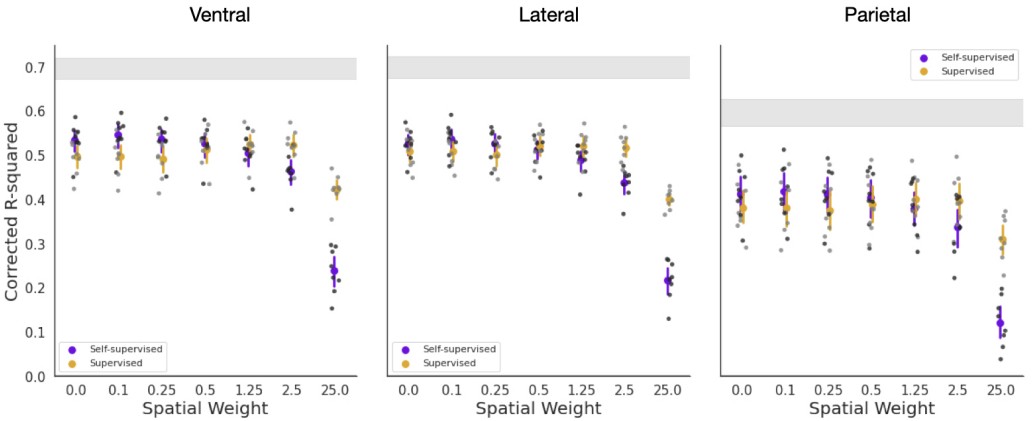

Fig. S12: Comparison of model fits for the topographic DCNNs across a range of spatial weights and the self-supervised vs. supervised task objectives. $R^2$ values are normalized by the noise ceiling (NC) of each voxel such that 1.0 corresponds to the intrinsic data NC. Each dot represents a subject. Shaded gray error bars: range of subject-to-subject NC.

## A.8 Smoothness of the mapping

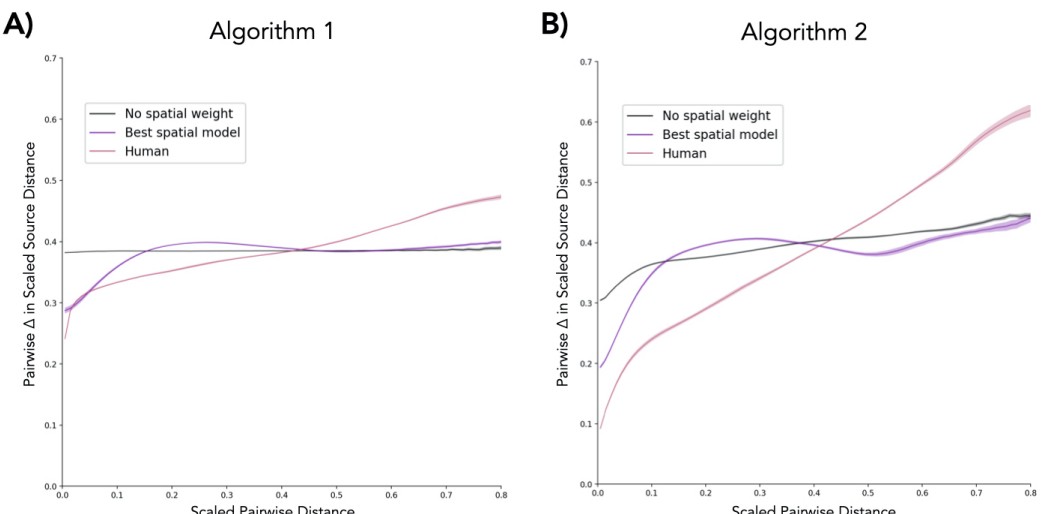

Fig. S13: How distance between assignments in the source space changes as distance between units in the target space increases for Algorithms 1 (A) and 2 (B). Best spatial model (purple) is $\alpha = 0.25$. All estimates are averaged across source subjects and the shaded region represents SE across source subjects.

## A.9   Additional limitations

A further limitation of this work is the static nature of the stimuli used in the neural data collection. The static nature of the stimuli and single task required of participants while viewing means we are limited to purely asking about the largely task-free generation of visual representations. It is entirely possible that the information carried by voxels in each stream would differ considerably given different task demands in data collection or a multimodal and dynamic dataset and we hope to see this question pursued in the future.

However, there is sufficient variability in responses across streams even within this static dataset, with highly reliable signal in Lateral and Parietal, as well as Ventral [14]. Furthermore, representations in Lateral and Parietal in this dataset have been shown to be distinct from each other, and from Ventral [14]. We hypothesize that this is due to the diverse and naturalistic nature of the images used in the experiment. The images contain much implied motion, which has been shown to reliably activate hMT+, with many scenes including activities such as as baseball games, cycling, and skiing [32]. These images, in combination with the many images containing social information, appear to be sufficient to drive the Lateral stream. The Parietal ROI has a slightly lower average noise ceiling and lower subject-to-subject reliability than the other two streams, but it is not clear that this would be due to a lack of dynamic stimuli, since hMT+ and other areas preferring dynamics and motion are actually located within the Lateral stream, which responds to these static stimuli as robustly as Ventral. The Parietal stream has instead been implicated in tasks such as object detection or spatial attention, which do not require motion. However, this stream might require different task demands, or it may be that due to different life experience, attention-related areas understandably differ in their response from subject to subject more than other areas. We believe this a fruitful avenue for future research but outside the scope of this study, for which static stimuli are sufficient.

