# OpenReview forum: "Topographic DCNNs trained on a single self-supervised task capture the functional organization of cortex into visual processing streams"
_NeurIPS.cc/2022/Workshop/SVRHM — SVRHM Oral_

### Official Review · Reviewer_N8mh · 2022-10-06
**A good use of topographic DCNN model to study functional organization of the brain**

**Rating:** 7
**Confidence:** 4

**Review:**

The authors used DCNN with topographic restriction during self-supervised training to build neural network model study the correspondence between artificial network unit and brain voxels. Interestingly, the authors show topographic DCNN also show different visual streams similar to human brain’s. The is a great showcase on how to use neural network models to shed light on how the brain develop to show such functional organizations by anatomical contraints.

Pros:
1.	Instead of linear regression, this paper shows a voxel to unit mapping, which is a novel method
2.	Using self-supervised learning provides a better insight on how different visual streams can be naturally emerged through contrastive learning
3.	This method provides a lot of opportunities to further model and understand the functional organization of the brain

Suggestions:
1.	The paper does not provide any performance metric of the model in performing any vision task (instead of the match between brains). I am wondering if the performance differs a lot with different spatial loss weighting
2.	In the method, two different tasks were mentioned to train the topographic DCNN. However, in the result session, the patterns between self-supervised and supervised task are not the same. However, the discussion between the results is not thorough. Readers would love to understand more about how the different training tasks would lead to different results.
3.	Typo in Figure 1 legend: missing “:” in the second last line.

---

> ### Author Response · Authors · 2022-12-21
> **Response to reviewer N8mh**
>
> We'd like to thank the reviewer for their comments. We respond to specific points below:
>
> *“The paper does not provide any performance metric of the model in performing any vision task (instead of the match between brains). I am wondering if the performance differs a lot with different spatial loss weighting.”*
>
> See response to reviewer emXY
>
> *“In the method, two different tasks were mentioned to train the topographic DCNN. However, in the result session, the patterns between self-supervised and supervised task are not the same. However, the discussion between the results is not thorough. Readers would love to understand more about how the different training tasks would lead to different results.”*
>
> Thanks for pointing this out. We've added more discussion about the different results for the supervised vs the self-supervised models to section 3.3 and plan to address this more fully in a future manuscript that is less space-limited.
>
> *“Typo in Figure 1 legend: missing “:” in the second last line.”*
>
> Fixed! Thank you

---

### Official Review · Reviewer_emXY · 2022-10-12
**A good paper on the emergence of visual streams**

**Rating:** 9
**Confidence:** 4

**Review:**

This paper proposes a model to explain how the organization of the visual system into three streams may emerge. Using self-supervised learning and a constraint encouraging local response correlations in a CNN and comparing its activities with the large scale NSD neuroimaging dataset, the authors present a model where different regions of a layer better match the activity of different visual streams. The paper is well-written, the analyses seem rigorous and the topic is timely. I recommend this contribution for SVRHM and provide some comments below in the spirit of helping improve the paper.


Comments:
1. In principle, the spatial loss is used to enforce similar activity patterns in neighbouring units. However, how it is computed is not clear to me. As far as I can tell, the authors correlate two vectors of size N_unit_pairs for each layer: C, which contains the correlations in activity patterns for each pair, and D, which contains the spatial distances between each pair. The spatial loss is low if these vectors correlate. This would mean that the loss would be low if large distance was correlated to similar activity patterns? This seems like the opposite of the intended purpose. I take it I misunderstood, but maybe the authors can clarify? In addition, it was not clear to me on which input data the loss is computed. Is it for each batch?

2. Typically, the ventral stream is modelled hierarchically, with different layers modelling different brain areas. Here, we only see the results for the last layer, mapped to large parts of visual cortex, which is quite different in spirit since there is no hierarchy, and it is not obvious how we should interpret what th earlier layers are doing. It would be interesting to see if different layers better explain different regions in terms of topography and activities, etc. It would also be interesting to see if removing the spatial loss decreases the match of the last layer with the brain's stream topography.

3.It would be interesting to report the performance of networks on their main objective for different values of alpha. Does alpha hurt performance? Or can it in fact help?

---

> ### Author Response · Authors · 2022-12-21
> **Response to reviewer emXY**
>
> We thank the reviewer for their kind words and address their comments below:
>
> *“In principle, the spatial loss is used to enforce similar activity patterns in neighbouring units. However, how it is computed is not clear to me. As far as I can tell, the authors correlate two vectors of size N_unit_pairs for each layer: C, which contains the correlations in activity patterns for each pair, and D, which contains the spatial distances between each pair. The spatial loss is low if these vectors correlate. This would mean that the loss would be low if large distance was correlated to similar activity patterns? This seems like the opposite of the intended purpose. I take it I misunderstood, but maybe the authors can clarify? In addition, it was not clear to me on which input data the loss is computed. Is it for each batch?”*
>
> We apologize for the confusion. D is actually the stabilized reciprocal Euclidean distance and so is instead more a measure of proximity. We have clarified this in the main text. The loss is computed for each batch and this clarification has also been added to the manuscript.
>
> *“Typically, the ventral stream is modelled hierarchically, with different layers modelling different brain areas. Here, we only see the results for the last layer, mapped to large parts of visual cortex, which is quite different in spirit since there is no hierarchy, and it is not obvious how we should interpret what the earlier layers are doing. It would be interesting to see if different layers better explain different regions in terms of topography and activities, etc.”*
>
> The topographic DCNN does model the visual hierarchy and we have confirmed that higher layers (e.g.the last conv layer of Resnet 18, layer 4.1) best correspond to higher-level visual areas across the streams and that earlier layers better map to early visual cortex. Here we show the results of mapping the units from the last layer to voxels in high-level visual areas, as this project as we are attempting to model only higher-level visual. However, we certainly agree that this is an interesting question! We plan to address this at length with regards to V1 in an upcoming paper (Margalit, Lee, Finzi, DiCarlo, Grill-Spector & Yamins, in prep), see also [11].
>
> *“It would also be interesting to see if removing the spatial loss decreases the match of the last layer with the brain's stream topography.”*
>
> We agree. Removing the spatial loss (which we can do simply by setting the weighting, $\alpha$, to 0) has a profound impact on the correspondence and greatly decreases the match of the last layer with both the brain's stream topography and the functional responses (Figures 3 & 4)
>
> *“It would be interesting to report the performance of networks on their main objective for different values of alpha. Does alpha hurt performance? Or can it in fact help?”*
>
> Object categorization, as measured by linear transfer performance to ImageNet top-1, is marginally lower for the topographic models at low spatial weights (median accuracy = 43.9%) vs models trained only on the self-supervised task objective (median accuracy = 48.5%). At the highest spatial weights, task performance declines considerably as expected. This will be discussed further in Margalit, Lee, Finzi, DiCarlo, Grill-Spector & Yamins, in prep. We are also interested in the question whether low alphas might actually help with transfer performance on some tasks and hope to investigate this further in the future.

---

### Official Review · Reviewer_a7xY · 2022-10-13
**Review of SVRHM paper on topographic DNNs**

**Rating:** 8
**Confidence:** 5

**Review:**

This paper compares the representations learned by topographic DNNs to high-level visual cortex, which here is split into three ROIs: ventral stream, lateral stream, dorsal stream. They find that ANNs trained with self-supervision self-organize in a topographic layer with different regions corresponding to each stream.

Overall, I like this paper, I think the methods are innovative (clever use of a smoothed Munkres algorithm for matching voxel to topographic network), and it’s an important subject, so I think it should be accepted. The criticisms which follow do not need to addressed for this conference, but if this turns into a full-fledged paper I would like to see this addressed:

1. The three-stream classification is a fairly new idea that hasn’t been adopted in the non-human primate and mouse literature and is still fairly controversial in the human literature. I think the authors will need, in a future paper, to figure out what is it about the representations that make them more lateral- or ventral- or dorsal-like, and confirm or infirm what is believed to be in each representation. They’ve tried without much success to do this with different tasks, and broadly speaking that didn’t work. They might want to try visualization of the high-level units, e.g. in the style of Olah et al.; these can be excellent diagnostic tools. They may also try to use probe stimuli, either from a bank of images or generated with DALL-E, Imagen, Stable Diffusion, etc.
2. As the authors point out in Appendix A.8, the stimuli are static images. This may be ok for ventral stream investigations, but it doesn’t make sense for dorsal and lateral stream investigations. The authors may be able to use existing datasets, such as the studyforrest dataset, which likely contains a sufficient number of objects, motion, and social relationships to map out these three streams.

---

> ### Author Response · Authors · 2022-12-21
> **Response to reviewer a7xY**
>
> We thank the reviewer for their comments and their suggestions to help strengthen our work. We respond to them in part here, and also plan to address them in further work.
>
> *“The three-stream classification is a fairly new idea that hasn’t been adopted in the non-human primate and mouse literature and is still fairly controversial in the human literature. I think the authors will need, in a future paper, to figure out what is it about the representations that make them more lateral- or ventral- or dorsal-like, and confirm or infirm what is believed to be in each representation. They’ve tried without much success to do this with different tasks, and broadly speaking that didn’t work. They might want to try visualization of the high-level units, e.g. in the style of Olah et al.; these can be excellent diagnostic tools. They may also try to use probe stimuli, either from a bank of images or generated with DALL-E, Imagen, Stable Diffusion, etc.”*
>
> The core results don't depend on acceptance of the three-stream classification, as we also observe stream clustering using purely the y-position of the voxels. However, this is probably not sufficiently explained in this paper given space constraints and we will make sure to expand upon this in future work. Note that we had similar concerns, and in prior work we examined the representational structure of the different streams using the NSD data. We found that the representational similarity structure in the ventral and lateral stream differ which provided additional empirical evidence for the present study (CCN 2022, https://2022.ccneuro.org/proceedings/0000598.pdf).
>
> We thank you for the suggestions. We will try such approaches in the future. In particular, one analysis we hope to run is to test readout for the model units assigned to the different streams to see if some units are better for some downstream tasks vs. others. While task doesn't appear to be a good organizing force, it is still possible (and frankly, probable, given known neuroscientific results) that the spatial constraints lead to functional differentiation. We would like to test if this is captured by our models and mappings.
>
> *“As the authors point out in Appendix A.8, the stimuli are static images. This may be ok for ventral stream investigations, but it doesn’t make sense for dorsal and lateral stream investigations. The authors may be able to use existing datasets, such as the studyforrest dataset, which likely contains a sufficient number of objects, motion, and social relationships to map out these three streams.”*
>
> We thank the reviewer for this comment. We believe this static dataset is sufficient for the purposes of this study (explanation and justification paragraph added to Appendix A.9) but agree that a naturalistic dynamic and multimodal dataset, potentially even with differing task demands, would be our ideal neural target. We plan to pursue this in future work and thank the reviewer for bring the studyforrest dataset to our attention, as we were not aware of this excellent resource.

---

### Official Review · Reviewer_TdCS · 2022-10-14
**Interesting findings**

**Rating:** 7
**Confidence:** 5

**Review:**

In this paper, the authors suggest that in ANNs trained with self-supervised learning and with topographic constraints, specialized modules and pathways similar to those in the visual system emerge. It is an intriguing finding and is consistent with a few other previous studies. The paper is very well written and I enjoyed reading it. I have a few comments that I explain below:

1- It is not intuitive how SimCLR can learn dorsal-like representations. Based on our current understanding of the ventral and the dorsal pathways (e.g. classic studies by Goodale and Milner), the main difference between the two pathways can be understood in terms of their representational invariances. For example, the ventral representations need to be viewpoint invariant (for e.g. object categorization), while the dorsal representations actually should be viewpoint dependent to properly guide actions, such as grasping. The augmentations used in SimCLR encourage learning viewpoint invariant representations; hence the good performance on object categorization. Because of these augmentations, it's hard to understand how dorsal-like representations could emerge in a model trained with SimCLR. Did the implemented SimCLR use the same augmentations in the original SimCLR paper? I am interested to hear the authors' explanation.

2- No measure of alignment between the ANNs and the brain, besides the topographic alignment, has been reported in the paper (a regression-based alignment for example). It'd be nice to see comparisons with an untrained ANN, an ANN only trained with the spatial correlation loss (very large $\alpha$), and also between the supervised and self-supervised ANNs. Is there any significant difference between the alignment scores of the three specialized streams?

3- The way the $\alpha$ value should be chosen isn't clear. In some of the plots, it seems that increasing $\alpha$ leads to an increase in the number of clusters, but the "correct" number of clusters is, in general, not known a priori. Why did the authors choose $\alpha=0.25$? Why the same $\alpha$ value was used for comparing the supervised and self-supervised models? One could imagine that a larger or smaller alpha value could potentially lead to better clustering with the supervised model. Also, unlike the ventral and lateral clusters, the parietal cluster isn't as clearly formed, which is also consistent with my first comment re the dorsal pathway.

---

> ### Author Response · Authors · 2022-12-21
> **Response to reviewer TdCS part 1**
>
> We'd like to thank reviewer TdCS for their comments. We respond to specific points below:
>
> *“It is not intuitive how SimCLR can learn dorsal-like representations. Based on our current understanding of the ventral and the dorsal pathways (e.g. classic studies by Goodale and Milner), the main difference between the two pathways can be understood in terms of their representational invariances ... The augmentations used in SimCLR encourage learning viewpoint invariant representations; hence the good performance on object categorization. Because of these augmentations, it's hard to understand how dorsal-like representations could emerge in a model trained with SimCLR. Did the implemented SimCLR use the same augmentations in the original SimCLR paper?”*
>
> SimCLR implemented in this work does use the same augmentations as in the original SimCLR paper, namely random Gaussian blur, random color distortions, random cropping (with resize back to the original size) followed by a random horizontal flip with 50% probability. Of these, only the random flip encourages viewpoint invariance, and only with 50% probability so there is still plenty of learning signal for crops of the same viewpoint, though the random cropping also encourages position invariance. However, it is important to note that augmentations used by SimCLR do not result in the complete loss of position and view information, but instead encourage the separable encoding ("detangling") of these representations. This is evidenced by findings that SimCLR-trained networks have high transfer accuracy to object pose detection, object position and object size tasks (Zhuang et al., 2021).
>
> Additionally, it's not clear that the ventral and dorsal streams should in fact be divided into viewpoint invariant and viewpoint dependent. The ventral stream contains regions that are both viewpoint-sensitive and viewpoint-invariant (Freiwald & Tsao, 2010), with reliable population receptive fields (pRFs) found in both ventral and lateral category-selective regions (Finzi et al., 2020), and position, size and pose can be reliably decoded from ventral cortex (Sayres 2008; Hong et al., 2016; Zoccolan et al., 2007). Furthermore, object-selective and viewpoint in,variant representations have been extensively documented in the dorsal stream, particularly in more posterior and medial regions (Sereno & Maunsell, 1998, Konen & Kastner, 2008, for review see Freud et al., 2016)
>
> *“No measure of alignment between the ANNs and the brain, besides the topographic alignment, has been reported in the paper (a regression-based alignment for example). It'd be nice to see comparisons with an untrained ANN, an ANN only trained with the spatial correlation loss (very large $\alpha$), and also between the supervised and self-supervised ANNs. Is there any significant difference between the alignment scores of the three specialized streams?”*
>
> We do include a measure of alignment in the manuscript, the correlations between model and brain responses from the one-to-one mappings (Figure 4E). However, we note that this is a novel method of alignment and thus understand the desire to also see a regression-based alignment. Consequently, we have added to the supplement results using Ridge regression to map from model responses to each of the three streams, for models with no spatial weight ($\alpha = 0$) and then a range of spatial weights up through $\alpha = 25$ for each of the supervised and self-supervised models (Figure S12). While we do find lower alignment scores for parietal vs ventral and lateral, parietal also has lower subject-to-subject alignment. We also note that traditional regression results are included in the supplement for a range of other more standard object categorization and action recognition trained networks (Figure S11)

---

> > ### Author Response · Authors · 2022-12-21
> > **Response to reviewer TdCS part 2**
> >
> > *“The way the $\alpha$ value should be chosen isn't clear. In some of the plots, it seems that increasing α leads to an increase in the number of clusters, but the "correct" number of clusters is, in general, not known a priori. Why did the authors choose α=0.25? Why the same α value was used for comparing the supervised and self-supervised models? One could imagine that a larger or smaller alpha value could potentially lead to better clustering with the supervised model.”*
> >
> > We chose to compare the self-supervised and supervised results using $\alpha = 0.25$ as the self-supervised $\alpha = 0.25$ model was the best over a combination of metrics. However, the reviewer makes a good point that a larger or smaller alpha value could lead to better clustering with the supervised model instead. To address this, we have now added results (Figures 4, S7, S8) for the supervised models at all values of alpha (range 0.0 to 25). While further increased spatial weighting does lead to more clustering and higher correlations for the supervised models, these models are still worse models of the three streams than their self-supervised counterparts, both quantitatively and qualitatively at all levels of alpha.
> >
> > As a final point, given a three-stream model of the brain, the "correct" number of clusters would be three. We acknowledge that there are other models of the brain and different views on how to draw boundaries between streams, and we have thus included the results where the mapping is visualized using the y-position of the chosen voxels (Figures 2 & 3). These results are agnostic to the number of streams or the ROI boundaries, and yet still seem to produce three-stream clustering.
> >
> > *“Also, unlike the ventral and lateral clusters, the parietal cluster isn't as clearly formed, which is also consistent with my first comment re the dorsal pathway.”*
> >
> > Yes, this is an important point. Interestingly, areas which many associate with the dorsal pathway, such as hMT+, are actually included in the lateral stream in humans, and the lateral stream appears to be well-modeled by the topographic DCNNs. However, we are clearly still missing something with regards to parietal. In addition to being less clearly formed, the parietal clusters always appear in the simulated cortex near ventral clusters, which is surprising given the actual anatomical order on human cortex (ventral then lateral than parietal). This seems to reflect a greater similarly of ventral and parietal representations (vs. lateral and parietal) in this dataset (Finzi et al., 2022) and suggests an area which requires further exploration. We'd like to speculatively mention that there is a white-matter fiber bundle called the vertical occipital fascicle (VOF) that directly connects ventral and parietal visual areas. We hope to explore the role of long-range connections in the future. We now briefly discuss this in section 4.